# Phytochemical Screening of *Himatanthus sucuuba* (Spruce) Woodson (Apocynaceae) Latex, In Vitro Cytotoxicity and Incision Wound Repair in Mice

**DOI:** 10.3390/plants10102197

**Published:** 2021-10-16

**Authors:** Oscar Herrera-Calderón, Lisbeth Lucia Calero-Armijos, Wilson Cardona-G, Angie Herrera-R, Gustavo Moreno, Majed A. Algarni, Mohammed Alqarni, Gaber El-Saber Batiha

**Affiliations:** 1Department of Pharmacology, Bromatology, Toxicology, Faculty of Pharmacy and Biochemistry, Universidad Nacional Mayor de San Marcos, Jr. Puno 1002, Lima 15001, Peru; lisluc21@gmail.com; 2Química de Plantas Colombianas, Faculty of Exact and Natural Sciences, Institute of Chemistry, University of Antioquia (UdeA), Calle 70 No. 52–21, Medellín 1226, Colombia; wilson.cardona1@udea.edu.co (W.C.-G.); angie.herrerar@udea.edu.co (A.H.-R.); gustavomorenoq@gmail.com (G.M.); 3Department of Clinical Pharmacy, College of Pharmacy, Taif University, P.O. Box 11099, Taif 21944, Saudi Arabia; m.alqarni@tu.edu.sa; 4Department of Pharmaceutical Chemistry, College of Pharmacy, Taif University, P.O. Box 11099, Taif 21944, Saudi Arabia; m.aalqarni@tu.edu.sa; 5Department of Pharmacology and Therapeutics, Faculty of Veterinary Medicine, Damanhour University, Damanhour 22515, Egypt; dr_gaber_batiha@vetmed.dmu.edu.eg

**Keywords:** latex, wound healing, anticancer, colon cancer, cytotoxicity, docking molecular

## Abstract

*Himatanthus sucuuba*, also known as “Bellaco caspi”, is a medicinal plant whose latex, stem bark, and leaves possess phenolic acids, lupeol, β-dihydro-plumbericinic acid, plumericin, and plumeride, among other components. Some of these have been linked to such biological activities as antiulcer, anti-inflammatory, and wound healing. The aim of this study was to determine the phytochemical compounds of *H. sucuuba* latex, as well as its in vitro cytotoxicity and wound healing effect in mice. Latex was collected in the province of Iquitos, Peru. Phytochemical analysis was carried out with UPLC-ESI-MS/MS. The cytotoxicity was evaluated on two colon tumor cell lines (SW480 and SW620) and non-malignant cells (human keratinocytes, HaCaT, and Chinese hamster ovary, CHO-K1). The mice were distributed into two groups, as follows: Group I—control (*n* = 10; without treatment); II—(*n* = 10) *H. sucuuba* latex; wounds were induced with a scalpel in the dorsal–cervical area and treatments were applied topically twice a day on the incision for 10 days. Molecular docking was carried out on the glycogen synthase kinase 3β protein. Twenty-four chemical compounds were determined, mainly flavonoid-type compounds. Latex did not have a cytotoxic effect on tumor cells with IC_50_ values of more than 500 µg/mL. The latex had a regenerative effect on wounds in mice. Acacetin-7-O-neohesperidoside had the best docking score of −9.9 kcal/mol. In conclusion, *H. sucuuba* latex had a wound healing effect in mice, as confirmed by histological study. However, a non-cytotoxic effect was observed on colon tumor cells SW480 and SW620.

## 1. Introduction

The Apocynaceae family comprises about 355 genera and 3700 species spread throughout the world, but mainly in tropical areas [1]. Within this family, the main genera are Rauwolfia, Catharanthus, Allamanda, Strophantus, and Himatanthus, which have been demonstrated to be useful for premature ejaculation, urinary tract infections, snake bites, fever, diarrhea, asthma, toothache, skin infections, and wound healing, among other things [2]. Himatanthus Wild. ex Schult. (Apocynaceae) is a genus comprising about 13 species of trees and shrubs, and these are widely distributed in Central and South America, particularly in Brazil [3]. Triterpenes, alkaloids, flavonoids, and iridoids are the most common chemical compounds found in its leaves, bark, and latex, and these have been shown to have anticancer, anti-inflammatory, and anthelmintic properties. Some species of the Himatanthus genus include *H. drasticus*, *H. phagedaenicus*, *H. attenuatus, H. obovatus, H. semilunatus*, *H. tarapotensis*, *H. revolutus*, *H. articulatus*, *H. bracteatus*, and *H. sucuuba* (Figure 1) [4,5].

*Himatanthus sucuuba* (Spruce) Woodson, known as “Bellaco Caspi”, is a rainforest tree that grows to between 8 and 16 m in height (Figure 1). Furthermore, its white flowers contain volatile compounds that give off a characteristic odor. The stem is 30–40 cm in diameter. Its leaves have a bright green color and are about 25–30 cm in length. Milky white latex is produced when any organ of the plant is fractured or injured. In some communities from the rainforest, latex is used to treat different skin abnormalities, such as ulcers, snake bites, skin wounds, lumbar pain, fever, gastritis, hernia, herpes, uterus inflammation, etc. [6]. Additionally, phytochemical studies of its leaves, stem bark, and latex have revealed the presence of β-dihydro-plumbericinic acid, plumericin, plumeride, uleine, vanillic acid, amyrin, cis-polyisoprene, fulvoplumierin, isoplumericin, iso-uleine, lupeol acetate, lupeol cinnamate, α-amyrin cinnamate [7,8], 5-demethylplumieride, and isoplumieride [9], some of which have antifungal, antibacterial, and cytotoxic activities [10].

In Peru, folkloric medicine sees medicinal plants used to ameliorate several illnesses and as adjuvants in treatments for pain, urinary infections, liver problems, obesity, gastric ulcers, diabetes, cancer, hypertension, inflammation, wound healing, etc. [11]. Despite some chemical compounds of *H. sucuuba* having been previously identified, the phytochemical compounds in Peruvian *H. sucuuba* latex have not yet been identified by chromatographic techniques. Therefore, our main aim was to identify the main compounds of *H. sucuuba* latex using an ultraperformance liquid chromatograph with mass–mass detector (UPLC-ESI-MS/MS) and demonstrate its cytotoxic effects on colon cancer cell lines, as well as incision wound repair in mice.

## 2. Results and Discussion

### 2.1. Identification of the Phytochemicals of H. sucuuba Latex by UPLC-ESI-MS/MS

In the latex of *H. sucuuba*, 24 compounds were detected. These were mainly C-flavonoids, four of which were unknown isomers of apigenin and chrysin, amino acids, alkaloids, and fatty acid amides. Compounds 14, 15, 17, and 18 correspond to new compounds, and to verify their structures, they should be isolated and analyzed using spectroscopic techniques (nuclear magnetic resonance of proton and carbon). Table 1 indicates the *m*/*z* values of the ions detected by full ESI-MS (positive and/or negative) and the fragments obtained by MS/MS for each of them; the error (in ppm) of the calculation is also indicated for the molecular formula (≤5 ppm). Figure 2 shows total ionic current chromatograms (TIC) for the samples, identifying the compounds via their respective retention times. Some compounds identified in the latex have been identified in other species of the Apocynaceae family, such as phenolic acid-like trans-4-coumaric acid and para-coumaric acid in *Carissa carandus* [12]. However, some of the tentative structures determined in the chromatographic analysis differed from other Himatanthus species, i.e., plumieride, an iridoid representative of this plant [13], lupeol, two isomers of amyrin, β-sitosterol, and proteins isolated from *Himatanthus drasticus* latex [5]. α-amyrin, lupeol, and lupeol acetate were also determined in the hexane fraction of *H. sucuuba* latex [14], and spirolactone iridoids such as plumericin and isoplumericin were identified in the ethanol extract of *H. sucuuba* stem bark [15]. Furthermore, in the non and polar fractions of the stem bark, iridoids such as plumeridoid C, plumericin, plumieridin, and allamandicin were also identified, as were other flavonoidal structures such as dihydrocajanin, naringenin, dalbergioidin, biochanin A, dihydrobiochanin A, and the lignan pinoresinol [8]. Finally, 2′-O-methylperlatolic acid was found in stem bark from Brazilian *H. sucuuba* [16].

Although many of the phytochemicals were tentatively identified, some of them have not been reported in the literature before, probably due to external factors such as the origin, environmental conditions, and ecosystem. This study constitutes the first major report carried out on this plant, which grows in the Peruvian rainforest.

### 2.2. Wound Healing of H. sucuuba Latex in Mice

Mice were treated with a pure concentrate of *H. sucuuba* latex without any previous dilution. The histological study (Figure 3) demonstrated the relationship between the phytochemical constituents and the histopathological changes observed in the wound healing effect, and several reports have concluded that the luteolin isolated from Martynia annua Linn has a wound healing effect in mice. The histological analysis showed well-organized collagen fibers and a high proliferation of fibroblast cells [17]. Likewise, luteolin administrated intraperitoneally at doses of 100 mg/Kg induced wound repair in diabetogenic rats, with increased fibroblasts, angiogenesis, and formation of collagen fibers at the wound site over the 14 days of evaluation [18]. The mechanisms involved may be explained by the fact that luteolin induces the proliferation of fibroblasts, which produce different types of collagens to form the extracellular matrix (ECM). Although luteolin seems to counteract wound healing due to the increase in fibronectin and the production of collagen I and collagen III, it also inhibits the activity of collagenase and hyaluronidase, generating a stable ECM. Keratinocyte migration helps to construct a cell sheet, promoting wound healing [19]. On the other hand, apigenin isolated from Helichrysum graveolens flowers has been shown to encourage wound healing, as well as having antioxidant and anti-inflammatory effects [20]. Apigenin is linked to reductions in oxidative stress, and it modulates the expression of inflammatory cytokines at both the transcriptional and post-transcriptional level [21]. The topical administration of apigenin (0.67 mg/Kg) reduces the acute allergic reaction of dermatitis; its main mechanism is to inhibit matrix metalloproteinase-1 expression, TNF-α gene expression, vascular cell adhesion molecule-1 (VCAM-1), and E-selectin [22]. Puerarin is also known as daidzein-8-C-hexoside and might be related to wound repair. In a study carried out in diabetic rats, the oral administration of puerarin produced reepithelization and angiogenesis [23]. Puerarin administrated in rats for 28 days produced bone regeneration [24]. Another compound named chrysin (5,7-Dihydroxyflavone) played a pivotal role in wound healing by reducing p53 and iNOS expression [25]. An in vitro study has suggested that in a high-glucose environment, chrysin can inhibit the phosphorylation of serine/threonine–protein kinases (AKT), extracellular signal-regulated kinase (ERK), and matrix metalloproteinase-2 (MMP-2), diminishing the effects of vascular endothelial growth factor (VEGF) and vascular endothelial growth factor (VEGFR).

In our work, we identified other isomers of chrysin, such as 8-C-pentosyl-6-C-hexosyl-chrysin, one alkaloid (lenticin), one amino acid (phenylalanine-betaine), and two fatty amides (9-palmitamide and octadecenamide), which may be involved in the anti-inflammatory and wound healing effects. In Figure 3, the topical application of latex had a regenerative effect on the wound area following 10 days of administration twice a day. As indicated by the histological changes, treatment with latex caused collagen formation and anti-inflammatory effects and closed injuries in less time compared to the control group without treatment. Our results are similar to those in a previous report, but these latter results were derived via 15 days of treatment with once-daily application [6].

### 2.3. Cytotoxic Activity of the H. sucuuba Latex

Tumor cell lines of colon cancer SW480 and SW620 were used in this study; here, the latex from *H. sucuuba* did not have any effect on these cells, with IC_50_ values above 500 µg/mL for each treatment (Table 2). The control treatment did not cause any morphological change or damage to cells after the treatment with 1% DMSO (Appendix A). To understand the mechanisms likely involved in the high resistance to the phytochemical constituents found in latex, it must be understood that cytokeratin tonofilaments (keratin proteins) are present in both cells (SW480 and SW620) as well as in the wound healing process [26]. A wide variety of keratins exist, but 16 are rapidly induced during the first step of keratinocytes’ wound response [27]. K16 is involved in epidermal hyper-proliferation and is highly expressed in wounded tissue [24]. The upregulation of K16 has been shown to participate in the wound healing of diabetic rats [28]. Although we did not evaluate this biochemical marker, it showed efficacy in reducing the wound healing time in the latex group compared to the control group; this suggests the phytochemical constituents of *H. sucuuba* might have a proliferative effect on colon cancer cells, which should be studied. An important distinction between wound healing and cancer is that the promigratory and hyperproliferative behaviors of keratinocytes are self-limiting; shortly after epithelium repair is completed, the keratinocytes return to their inactive state. Hence, a loss of control over cell migration and proliferation leads to cancer [29]. Another potentially relevant mechanism is the relationship between flavonoids and antioxidant activity, which would have a significant effect on cytotoxicity due to the imbalance in the reduction–oxidation system. Therefore, the antioxidant activity of these compounds could have a protective effect on colon cancer, which was not shown with 5-FU, a drug that induces oxidative stress in tumor cells. Marullo et al. [30] determined that exposure to cisplatin (a commercial antitumoral drug) induces a mitochondrial-dependent reactive oxygen species (ROS) response that enhances the cytotoxic effect of nuclear DNA damage in tumor cells. Although latex lacked cytotoxic effectivity on colon cancer cells, in vivo studies are necessary to confirm our in vitro results because isolated compounds such as luteolin [31] and apigenin [32] have been demonstrated to have an antitumoral effect on colon cancer induced in mice.

The high IC_50_ values of HaCat (human keratinocytes) and CHO-K1 cells (Chinese hamster ovary) demonstrate that latex compounds did not cause any toxic damage. A study showed that luteolin inhibits the TNF-induced production of IL-6 and IL-8 and VEGF proliferation in HaCat cells [33]. Additionally, apigenin enhances UVB-induced apoptosis in human keratinocytes through both the extrinsic and intrinsic apoptotic pathways [34]. Polyphenols might have an effect on cell protection according to Lombardi et al. [35]; quercetin and myricetin had a cytoprotective effect on CHO-K1 cells exposed to enniatins. Puerarin also inhibited the secretion of inflammatory cytokines and chemokines in HaCat cells [36]. Regarding index selectivity, 5-FU was less selective than latex (SI < 1), but showed high cytotoxicity on colon cancer cell lines. Numerous chemical compounds of natural origin are involved in cytotoxicity, and possibly in anticancer effects. However, *H. sucuuba* latex contains varieties of phytoconstituents that might synergize or antagonize its biological activity; depending on the content percentage of each secondary metabolite, these could act via multiple mechanisms of cell protection, including for colon tumor cells [37,38].

### 2.4. In-Silico Study of the Main Components Determined in H. sucuuba Latex on Glycogen Synthase Kinase 3β Protein (PDB IDs: 1Q5K)

In this study, we selected the glycogen synthase kinase 3 (GSK3), which is a serine/threonine kinase with two isoforms, α and β. GSK3 has been associated with pathological conditions such as inflammation, diabetes, colon cancer, wounds, and Alzheimer’s disease [39,40]. The inhibition of GSK3 promotes wound healing through the β-catenin-dependent Wnt signaling pathway, triggering the proliferative phase and initiating the re-epithelialization of the wound, angiogenesis, the formation of the extracellular matrix, and the proliferation of keratinocytes and fibroblasts [41]. In this way, phytochemical constituents were docked against this regulatory enzyme, and these could act as potential inhibitors, enhancing the wound healing effect.

Molecular docking studies were performed to decipher the binding aspects of ligands (trans-4-coumaric acid, phenylalanine betaine, acacetin-7-O-neohesperidoside, lenticin, puerarin, luteolin, apigenin, chrysin, palmitamide, and 9-octadecenamide) and glycogen synthase kinase 3 protein (PDB IDs: 1Q5K). The images of docked complexes and molecular surfaces, as well as 3D and 2D interactive plots for ligands with the glycogen synthase kinase 3 protein, are shown in Figure 4 and Appendix A Molecular docking studies revealed that the highest binding affinity for the glycogen synthase kinase 3 protein was exhibited by acacetin-7-O-neohesperidoside, with the lowest binding energy (ΔG) of −9.9 kcal/mol and a predicted inhibitory concentration (Ki) of 1.6 nM. The ligand acacetin-7-O-neohesperidoside has been shown to be involved in conventional hydrogen bonding with Val135 and hydrophobic interactions with Gly63, Try134, Pro136, Thr138, and Arg141. In addition, other non-bonded interactions were observed, such as pi-alkyl, alkyl, and pi-sulfur interactions (Figure 4). On the other hand, luteolin displayed a significant affinity for glycogen synthase kinase 3, with a binding energy of −8.6 kcal/mol and Ki of 5.2 nM. Here, a conventional hydrogen bond was formed with Lys85 residue at the binding cavity of the glycogen synthase kinase 3 protein. Apigenin also showed a high binding energy of −8.5 and a KI of 5.7, with hydrogen bonds formed with Lys183 residue. The other ligand interactions with their binding energy and Ki are shown in Table 3.

The molecular docking studies show the possible role of acacetin-7-O-neohesperidoside potentiates as an inhibitor, with significant binding energy and predicted inhibitory concentration. This result is comparable with earlier reports on chlorogenic acid, ferulic, and caffeic acid, which also showed greater affinity for glycogen synthase kinase 3. It can be suggested that acacetin-7-O-neohesperidoside might exhibit similar bioactivity to chlorogenic, ferulic, and caffeic acid [42]. In our study, acacetin-7-O-neohesperidoside has been identified as the main metabolite associated with the wound healing effect. However, *H. sucuuba* latex also contained other potential compounds that could act in wound repair, such as puerarin, lenticin, and chrysin (Appendix A).

## 3. Materials and Methods

### 3.1. Chemicals and Reagents

Methanol, ethanol, acetonitrile, and formic acid (HPLC grade solvents) were acquired from Merck (Lima, Peru). Sulforhodamine B (SRB), Dulbecco’s modified eagle medium (DMEM), streptomycin, and dimethyl sulfoxide (DMSO) were purchased from Sigma-Aldrich Corporation, St. Louis, MO, USA.

### 3.2. Cell Lines and Culture Medium

Colon tumor cell lines and normal cells were used to evaluate the cytotoxic effect (colon cancer cell line (SW480), its metastatic derivative (SW620), and non-malignant cells (human keratinocytes, HaCaT, and Chinese hamster ovary, CHO-K1)). Cell lines were purchased from the European Collection of Authenticated Cell Cultures (ECACC), England. They were maintained in Dulbecco’s modified eagle medium (DMEM), supplemented with 10% heat-inactivated horse serum. To avoid bacterial contamination, 1% penicillin/streptomycin was used together with 1% non-essential amino acids (Gibco Invitrogen, Carlsbad, CA, USA). Additional supplements of 3% horse serum medium with 5 ng/mL selenium, 5 mg/mL transferrin, and 10 mg/mL insulin (ITS-defined medium; Gibco, Invitrogen, Carlsbad, CA, USA) were used.

### 3.3. Experimental Animals

Twenty Balb/C mice (8 weeks old, male, body weight 25–30 g) were purchased from the Bioterio of the Centro Nacional de Productos Biológicos, INS (Lima, Peru). Mice received standard pelletized food and drinking water *ad libitum*, and were acclimatized on a 12 h light/dark cycle before and during the study. This experimental protocol was enforced according to the international guidelines for experimental animals [43]. The protocol was approved by the institutional committee (Id: 206/FFB-UPG/2019). Animals were euthanized with ketamine/xylazine anesthesia, and unnecessary stress was avoided during the experiment.

### 3.4. Collection of Latex of H. sucuuba

Latex (500 mL) was collected from wild species by making a longitudinal incision in the stem. The collection location was Iquitos (10°00′13” S, 76°12′17” W), department of Loreto, Peru, in February 2019. Then, it was transported to the laboratory and stored in an amber flask at 4 °C. Samples of leaves and flowers were also selected for taxonomic identification at the Natural History Museum of the Universidad Nacional Mayor de San Marcos (UNMSM). The botanical identification code No. 010- USM-2019 was deposited for further studies.

### 3.5. Chromatographic Analysis

#### 3.5.1. Sample Preparation

Three grams of latex were put into a distillation flask and reduced until dryness with a Buchi Rotavapor R-200, Switzerland (40 °C, 130 mbar). Finally, we obtained 0.5 grams of creamy dark solid. Then, 30 mg was weighed and extracted with methanol:water (80:20) using an ultrasonicator (40 kHz, heat power 150 W; Branson 3800, MO, USA) for 30 min. The resulting solution was filtered through a 0.2 μm membrane disc filter and injected into the chromatographic system.

#### 3.5.2. UPLC-ESI-MS/MS Analysis of the Latex of *H. sucuuba*

The analysis was carried out on a Dionex Ultimate 3000 UHPLC System (Thermo Scientific, Bremen, Germany) triple quadrupole instrument and a mass spectrometer (Q Exactive Plus (Thermo Scientific, Bremen, Germany)) with a Luna© Omega C18 100 Å column, Phenomenex (150 × 2.1 mm, 1.6 μm), column temperature 30 °C, and a 0.3 m/mL flow rate. The mobile phase comprised 1% formic acid (A) and acidified methanol containing 1% formic acid (B). The elution conditions were 0–1 min, isocratic elution at 90% A and 10% B; 1–25 min, linear gradient from 5% A to 95% B; 0–1 min, linear gradient from 5% A to 95% B; 1–3 min, linear gradient from 90% A to 10% B; and 3–6 min, isocratic elution at 90% A and 10% B. The ionization source parameters were set using a positive and negative ion mode as follows: spray voltage 3.5/2.5 KV; capillary temperature 260 °C; gas carrier N2 (sheath gas flow rate 48, sweep gas flow rate 1); gas heater temperature 300 °C; S-lens RF level 100; normalized collision energy 30. Full MS scan parameters: range 120–1500 *m*/*z*; resolution 35,000; microscans 1; AGC target 5 × 106; maximum IT 80 ms. MS^2^ parameters: resolution 17,500; AGC target 1 × 10^6^; maximum IT 100 ms [44].

### 3.6. Evaluation of the Wound Healing Effect in Mice

Mice were grouped as follows. Group 1 (*n* = 10) consisted of distilled water as a placebo control, and group 2 (*n* = 10) was treated with 1.0 mL of latex from *H. sucuuba* in its natural form, without any dilution, according to its ethnopharmacological use. Treatments were administrated topically for 10 days twice a day (9:00 a.m. and 5:00 p.m.). In this study, mice were anesthetized to make an incision of 1 cm on their shaved back. The progression of wound healing was recorded at 24 h and at 7 and 10 days. On day 10, mice were euthanized, and tissue cuts of the treated zone were immediately fixed in buffer formol, embedded in paraffin, sliced into sections 4 μm thick, and stained with hematoxylin and eosin (H&E). Finally, the samples were analyzed using an optic microscope (Olympus BX51; Olympus Corporation, Tokyo, Japan) [6].

### 3.7. Cytotoxic Activity

The cytotoxicity of latex and 5-FU was assessed using the SRB cytotoxicity test. In 96-well tissue culture plates, cells were seeded to a final density of 20,000 cells/well and incubated at 37 °C in a humidified atmosphere with 5% CO_2_. After allowing the cultures to grow for 24 h, the cells were treated with 1% DMSO (control), 5-FU, or latex of *H. sucuuba* at escalating concentrations (5–1000 µg/mL). After treatment, the cells were fixed for one hour at 4 °C with trichloroacetic acid (50% *v*/*v*). After staining with 0.4% SRB to assess cell proteins, the plates were rinsed with 1% acetic acid to remove unbound SRB. Protein-bound SRB was solubilized in 10 mM Tris-base, then the absorbance was measured at 492 nm in a microplate reader (Mindray MR-96A, Shenzhen, China). All experiments assessing the cytotoxicity effect were performed in triplicate [45]. The selectivity index (SI) was calculated to determine the cytotoxic selectivity of the evaluated substances based on the following formula: IC_50_ of the normal cells (HaCaT and CHO-K1)/IC_50_ of the tumor cells (SW480 and SW620). If SI is more than 1, the substance was more cytotoxic to tumor cells than normal cells.

### 3.8. Molecular Docking of Glycogen Synthase Kinase 3β and Ligands

The intention of this study is to elucidate the binding propensities of ligands trans-4-coumaric acid, phenylalanine betaine, acacetin-7-O-neohesperidoside, lenticin, puerarin, luteolin, apigenin, chrysin, palmitamide, 9-octadecenamide, and glycogen synthase kinase 3 protein (PDB IDs: 1Q5K). Before the docking study with the phytochemicals of *H. sucuuba*, protein files were retrieved from the PubChem database and protein data bank, respectively. Before performing molecular interaction studies, glycogen synthase kinase 3β [42] was further assessed for its missing side-chain residues using the openMM simulation tool (https://openmm.org/ accessed on 6 September 2021). Molecular docking studies were performed using Autodock v4.2.6. The binding cavity for the ligands in glycogen synthase kinase 3 proteins was determined from the predefined co-crystallized X-ray structure of RCSB PDB. The residue positions were calculated within 3 Å from the co-crystallized ligand. After cavity selection in each case, the co-crystallized ligands were removed using the Chimera tool (https://www.cgl.ucsf.edu/chimera/ accessed on 6 September 2021), and, subsequently, the energy was minimized using the steepest descent and conjugate gradient algorithm. Finally, merging the nonpolar hydrogens, both receptor and target compounds were saved in the pdbqt format. Grid boxes were created with specific dimensions in a 0.3 Å space. Following the Lamarckian genetic algorithm (LGA), docking studies of the protein–ligand complex were performed to determine the lowest free energy of binding (∆G). The molecular docking studies were performed in three replicates, with a total of 50 solutions computed in each case, and with a population size of 500, number of evaluations of 2,500,000, and a maximum number of generations of 27,000; the rest of the default parameters were used. After docking, RMSD clustering maps were obtained by re-clustering with clustering tolerances of 0.25 Å, 0.5 Å, and 1 Å, respectively, to obtain the best cluster with the lowest energy score and a large population.

### 3.9. Statistical Treatment

The half inhibitory concentration (IC_50_) was determined via linear regression analysis. Spearman’s rho correlation coefficients (r) were determined, which indicate the relationship between the concentrations and the growth percentage. Microsoft Office Excel 2016 was used to create the database and perform the statistical calculations.

## 4. Conclusions

Based on our findings, latex from *H. sucuuba* contains phytochemicals (mainly of a flavonoidal structure) that could be linked to a wound healing effect. Administration twice a day produced a regenerative effect, which was observed in our histopathological study in mice skin wounds. Moreover, *H. sucuuba* latex did not have any cytotoxic effect on colon cancer cell lines SW480 and SW620, and was non-toxic for HaCaT and CHO-K1 cells. In addition, to postulate the mechanisms of the phytochemicals found in latex, molecular docking was carried out on glycogen synthase kinase 3β (PDB IDs: 1Q5K), a regulatory enzyme active in wound healing, the inhibitors of which enhance this process. From the ten evaluated phytochemicals, acacetin-7-O-neohesperidoside had the best docking score of −9.9 kcal-mol^−1^, followed by luteolin and apigenin. However, the wound healing effect might be a synergistic consequence of all its components. Further studies are necessary to find any relationship between the in silico and in vivo findings. Likewise, animal models of colon cancer should be tested to analyze and confirm our results in vitro.

## Figures and Tables

**Figure 1 plants-10-02197-f001:**
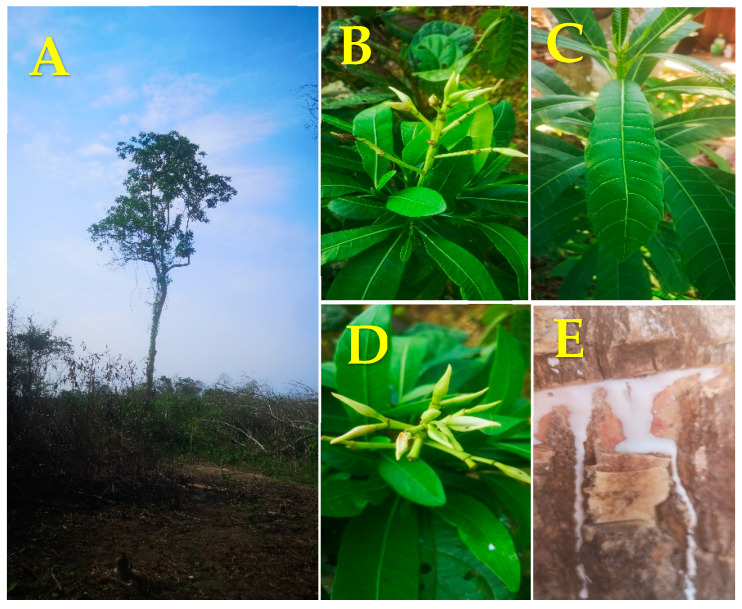
Photographs of Bellaco caspi (*Himatanthus sucuuba* (Spruce) Woodson). (**A**) Adult tree; (**B**–**D**) Young inflorescences; (**C**) Leaves; (**E**) Latex from the cut stem.

**Figure 2 plants-10-02197-f002:**
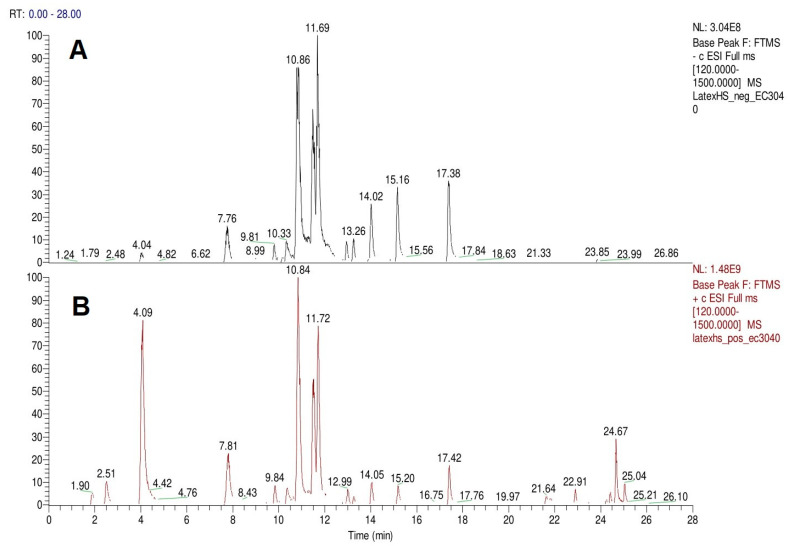
TIC chromatograms in negative (**A**) and positive (**B**) ESI mode of *H. sucuuba* latex.

**Figure 3 plants-10-02197-f003:**
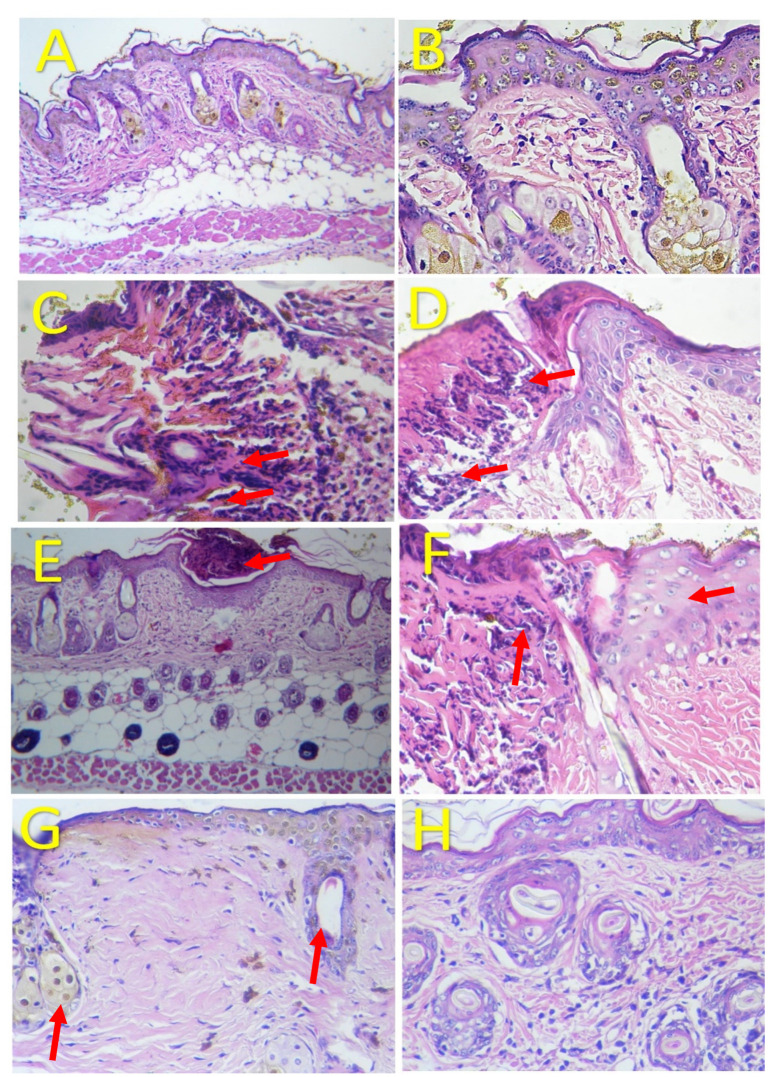
Histological analysis of skin tissue healing. Photomicrographs of (**A**–**H**): Hematoxylin and Eosin (H&E) staining with 10× and 40× magnification. (**A**,**B**): (Control group). The epithelial surface, the connective tissue underneath, the hair follicles, and the sebaceous glands are unaltered. (**C**): (Control group; 24 h): A loss of epithelium and the infiltration of polymorphonucleate leukocytes are clear. The epithelium also presents bleeding and fibrinoid exudate. (**D**): (Control group; 7 days): This shows an inflammatory reaction to polymorphonuclear leukocytes. (**E**): (Control group; 10 days): A small superficial abscess can be observed in the outermost layer of the epidermis. (**F**): (Latex group; 24 h): This reveals a dense inflammatory reaction and the presence of polymorphonuclear leukocytes. However, the epithelium is better preserved, with keratinocytes present below the superficial infiltrate. (**G**): (Latex group; 7 days): An area with fewer insertions (hairs and sebaceous glands), with more collagen fibers ranging from dense to thick. (**H**). (Control group; 10 days): Here, the skin is normal.

**Figure 4 plants-10-02197-f004:**
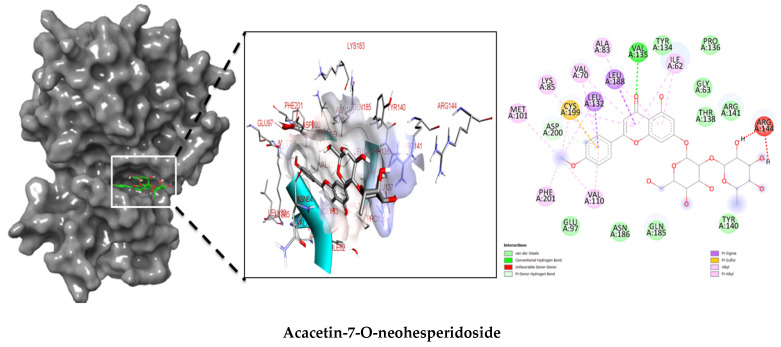
Molecular interaction studies of phytochemical constituents of *H. sucuuba* latex (Acacetin-7-O-neohesperidoside, luteolin, and apigenin) with glycogen synthase kinase 3-β (PDB IDs: 1Q5K), surface view (Left panel), and 2D (Right panel) interactions.

**Table 1 plants-10-02197-t001:** Tentative chemical components identified by UPLC-ESI-MS/MS of *H. sucuuba* latex.

**N°**	**Rt**	**MS-ES^-^**	**MS^2^**	**Error** **(ppm)**	**MS-ES^+^**	**MS^2^**	**Error** **(ppm)**	**MF/MM**	**Chemical Structure**
**1**	2.49	163.0	119.1 93.0	2.14	−	−	−	C_9_H_8_O_3_ 164.16	trans-4-Coumaric acid 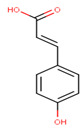
**2**	4.09	−	−	−	208.1	149.1 131.1 107.1 103.1 60.1	−0.65	C_12_H_17_NO_2_ 207.27	Phenylalanine betaine 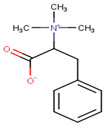
**3**	7.76	−	−	−	247.1	188.1 146.1 144.1 118.1 98.9 60.1	−0.76	C_14_H_18_N_2_O_2_ 246.30	Lenticin 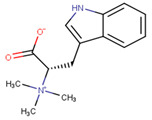
**4**	9.77	593.2	519.1	−0.01	595.2	457.1 439.1 421.1 409.1 379.1 349.1 337.1 325.1 307.1 295.1	−2.12	C_27_H_30_O_15_ 594.50	Apigenin 6,8-dihexoside (Vicenin II) 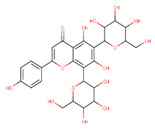
**5**	9.81	591.17	559.2 487.1 469.1 439.1 395.1 367.1 335.1 307.1	1.65	593.2	539.2 513.1 437.1 419.1 393.1 369.1 339.1	−0.34	C_28_H_32_O_14_ 592.55	Unknown 1 (isomer 1) Tentative structure: Acacetin 7- neohesperidoside 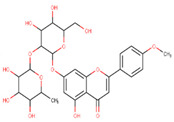
**6**	9.96	577.2	545.1 485.1 459.1 429.1 395.1 365.1 335.1 247.1	1.74	579.2	483.1 459.1 429.1 423.1 393.1 369.1 339.1	−0.09	C_27_H_30_O_14_ 578.53	6-C-Hexosyl-8-C-deoxyhexosyl-apigenin (isomer 1) 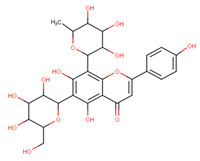
**7**	10.16	563.1	545.1 503.1 473.1 443.1 425.1 383.1 353.1 325.1	1.51	565.2	427.1 409.1 391.1 379.1 361.1 349.1 337.1 325.1 295.1	0.05	C_26_H_28_O_14_ 564.50	8-C-Hexosyl-6-C-Pentosyl-apigenin (isomer 1) 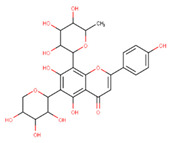
**8**	10.33	593.2	519.1 489.1 471.1 429.1 399.1 369.1 339.1	0.09	595.2	523.1 481.1 463.1 427.1 409.1 365.1 353.1 341.1 323.1	−2.21	C_27_H_30_O_15_ 594.53	8-C-Hexosyl-6-C-hexosyl-apigenin 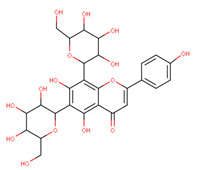
**9**	10.55	563.1	545.1 503.1 473.1 443.1 425.1 383.1 353.1 325.1	0.97	565.2	433.1 409.1 391.1 379.1 361.1 349.1 337.1 325.1 295.1	−0.69	C_26_H_28_O_14_ 564.50	6-C-Hexosyl-8-C-pentosyl-apigenin (isomer 2) 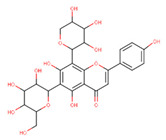
**10**	10.6	591.2	559.1 487.1 469.1 455.1 395.1 365.1 335.1 307.1	1.45	593.2	539.2 513.1 437.1 419.1 393.1 369.1 339.1	−0.31	C_28_H_32_O_14_ 592.55	Unknown 1 (isomer 2)
**11**	10.84	577.2	487.1 457.1 413.1 383.1 353.1 325.1 191.0	1.21	579.2	447.1 423.1 405.1 393.1 379.1 349.1 337.1	−0.31	C_27_H_30_O_14_ 578.53	6-C-Hexosyl-8-C-deoxyhexosyl-apigenin (Isomer 2) 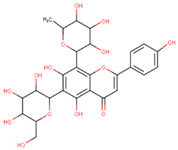
**12**	10.95	547.1	529.1 487.1 457.1 427.1 409.1 367.1 337.1 309.1	0.96	549.2	307.1 393.1 375.1 363.1 333.1 321.1 309.1 291.1 279.1	−0.76	C_26_H_28_O_13_ 548.50	8-C-Pentosyl-6-C-hexosyl-chrysin (Isomer 1) 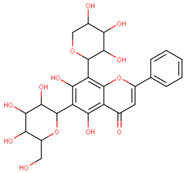
**13**	11.31	547.1	529.1 487.1 457.1 427.1 397.1 367.1 337.1 309.1	0.85	549.2	393.1 375.1 363.1 333.1 321.1 309.1 291.1 279.1	−0.43	C_26_H_28_O_13_ 548.50	8-C-Pentosyl-6-C-hexosyl-chrysin (Isomer 2) 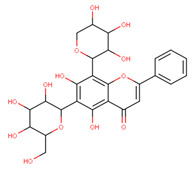
**14**	11.5	561.2	487.1 457.1 441.1 423.1 367.1 337.1 309.1	1.11	563.2	407.1 389.1 371.1 363.1 339.01 321.1 309.1 291.1 279.1	−0.46	C_27_H_30_O_13_ 562.53	Unknown 2 (Isomer 1) (Tentative structure) 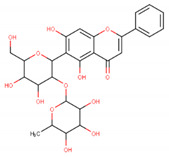
**15**	11.69	561.2	471.1 457.1 441.1 397.1 367.1 337.1 309.1 191.0	0.66	563.2	431.1 387.1 377.1 363.1 333.1 321.1 309.1 291.1	−0.35	C_27_H_30_O_13_ 562.53	Unknown 2 (Isomer 2)
**16**	12.83	543.2	469.1 453.1 423.1 379.1 361.1 349.1 319.1 95.1	1.16	545.2	455.1 431.1 389.1 377.1 359.1 333.1 321.1 309.1 279.1	0.22	C_27_H_28_O_12_ 544.51	Unknown 3 (C-flavonoid)
**17**	12.98	385.1	325.1 307.1 295.1 267.1	1.83	387.1	369.1 351.1 327.1 321.1 297.1 267.1 205.0 105.0	−0.45	C_20_H_18_O_8_ 386.36	Unknown 4 (Isomer 1) Tentative structures 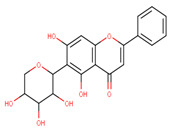
**18**	13.26	385.1	325.1 307.1 295.1 267.1 253.1	2.06	387.1	369.1 351.1 333.1 321.1 305.1 297.1 281.1 267.1 105.0	−0.37	C_20_H_18_O_8_ 386.36	Unknown 4 (Isomer 2)
**19**	14.02	415.1	399.1 381.1 363.1 337.1 313.1 283.1 219.1	−0.5	417.1	341.1 311.1 283.1 269.0	1.47	C_21_H_20_O_9_ 416.40	Daidzein-8-C-hexoside (Puerarin) 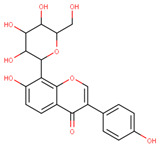
**20**	14.1	285.0	175.0 151.0 133.0	2.37	287.1	153.0	−1.13	C_15_H_10_O_6_ 286.24	Luteolin 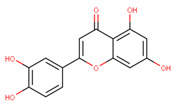
**21**	15.16	269.0	225.1 201.1 181.1 151.0 149.0 117.0	3.01	271.1	153.0 119.0	−0.96	C_15_H_10_O_5_ 270.24	Apigenin 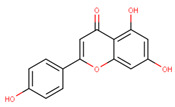
**22**	17.38	253.1	209.1 181.1 143.0 107.0 63.0	3.18	255.1	209.1 171.0 153.0 129.0 67.0	−1.12	C_15_H_10_O_4_ 254.24	Chrysin 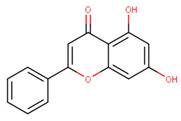
**23**	24.41	−	−	−	256.3	130.1 116.1 102.1 88.1 74.1	−0.59	C_16_H_33_NO 255.24	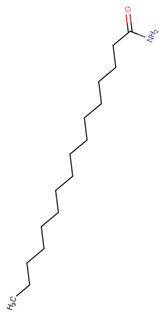
**24**	24.67	−	−	−	282.3	265.3 247.2 177.2 149.1 135.1 121.1 97.1 83.1 69.1	−0.64	C_18_H_35_NO 281.50	9-Octadecenamide 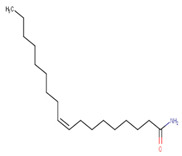

Rt: retention time; MF/MM: molecular formula/molecular mass.

**Table 2 plants-10-02197-t002:** Cytotoxic activity of *H. sucuuba* latex on colon cancer cell lines.

**Treatment**	**24 Hours**							
	**IC_50_ µg/mL**				**Selectivity Index (SI)**			
	**SW480**	**SW620**	**HaCat**	**CHO-K1**	**HaCat/SW480**	**CHO-K1/SW480**	**HaCat/** **SW620**	**CHO-K1/** **SW620**
Control	NI	NI	NI	NI	−	−	−	−
*H. sucuuba*	NI	>500	>500	>500	>1	>1	1	1
5-FU	200.84 ± 17.30	116.91 ± 12.00	50.84 ± 17.41	70.69 ± 5.15	0.25	0.35	0.43	0.60
**Treatment**	**48 Hours**							
	**IC_50_ µg/mL**				**Selectivity Index (SI)**			
Control	NI	NI	NI	NI	−	−	−	−
*H. sucuuba*	NI	>500	>500	314.7 ± 16.42	>1	<1	1	<1
5-FU	22.67 ± 1.35	23.53 ± 2.12	20.16 ± 1.01	22.53 ± 1.41	0.89	0.99	0.86	0.96

NI: non-inhibition; 5-FU: 5-fluorouracil. SI: values more than 1 are considered very selective for tumor cells.

**Table 3 plants-10-02197-t003:** Binding energy, predicted inhibitory concentration profile, and residue interactions between ligands and glycogen synthase kinase 3β (PDB IDs: 1Q5K).

Complex	Binding Energy (ΔG) (kcal/mol)	Ki (nM)	Polar Contact Interactions	Non-Bonded Interactions
trans-4-Coumaric acid	−6.0	51	Val135, Tyr134, Ala83, val110, Leu132, Asp133, Leu188, Asp200, Gly65	Val70, Cys199, Lys85
Phenylalanine betaine	−5.6	73	Lys85, Arg96, Glu97, Asn95, Gly202	Phe67, Val87
Acacetin-7-O-neohesperidoside	−9.9	1.6	Glu97, Asn186, Gln185, Tyr140, Val135, Tyr134, Asp200, Pro136, Gly63, Thr138, Arg141	Arg144, Ile62, Leu188, Leu132, Cys199, Phe201, Val110, Met101, Lys85, Val70
Lenticin	−6.5	33	Asp200, Leu132, Tyr134, Gly63, Gln185, Thr138, Ile62, Arg141	Val70, Cys199, Lys85, Leu188, Ala83
Puerarin	−8.3	6.8	Asn64, Gly65, Ser66, Lys85, leu132, val110, Tyr134, Ile62, Gln185, Asn186, Lys183, Gly63	Asp133, leu188, Val135, Val70, Asp200, Cys199
Luteolin	−8.6	5.2	Tyr134, Asp133, Leu132, Val110, Lys85, Asp200, Gly65, Asn64, Gly63	Val135, Ala83, leu188, Val70, Cys199, Ile62
Apigenin	−8.5	5.7	Phe67, Ser66, Lys183, Gln185, Asn186, Leu188, Val110, Leu132, Ala83, Gly65, Phe67, Ser66	Asp200, Val70, Cys199, Lys85
Chrysin	−8.2	7.4	Gly65, Asp200, Lys85, Leu132, Val110, Asp133, Val135, Tyr134	Cys199, Ile62, Ala83, Leu188, Val70
Palmitamide	−5.6	73	Gly68, Val110, Val135, Phe67, Ser66, Gly202, Asp200	Leu188, Lys199, Val70, Ala83, Leu132, Ile62, Lys85, Lys199, Tyr134
9-Octadecenamide	−5.7	67	Gly65, Phe67, Asp200, Ser66, Ser203, Asp181, Gly202	Ile62, Leu188, Leu132, Ala83, Val70, Cys199, Lys85

## Data Availability

The data that support the findings of this study are available from the corresponding author upon reasonable request.

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
