# Peer review of "Phytochemical Screening of Himatanthus sucuuba (Spruce) Woodson (Apocynaceae) Latex, In Vitro Cytotoxicity and Incision Wound Repair in Mice"

_plants, 2021, doi:10.3390/plants10102197_

Round 1
Reviewer 1 Report
The article provides new information on the composition of phytochemicals of the milky sap (latex) of the species Himatanthus sucuuba ( Spruce ), as well as an experiment on the effect of latex doses on cytotoxicity and the possibility of wound healing in mice. The UPLC-ECI- MS / MS technique also identifies compounds not previously identified in other literatures, providing important information on how the potential site, i.e. the specific pedoclimatic conditions in which the plants grew (Peru), play a role in phytochemicals composition in plant material, latex.
The introduction is clearly written and provides all the necessary facts about the research questions. The results and discussion are presented and explained clearly and concisely, and all tables and figures are necessary to present the results adequately. The conclusion is in line with the aim of the paper and clearly highlights the main findings, but also provides a critical review of the research that should be carried out as a follow-up.
In the materials of the paper, it is necessary to describe in more detail the origin of the plant material, but I give all the suggestions below.
Precisely because of all these points, after minor corrections according to the suggestions, I recommend the publication of the paper.
TITLE - I suggest a shorter, more concrete title, for example: Phytochemical screening of Himatanthus sucuuba (Spruce) Woodson (Apocynaceae) latex, in vitro cytotoxicity and incision wound repair in mice.
ABSTRACT
Lines 21-22 - I propose for the first sentence a brief description of the phytochemicals of Himatanthus sucuuba and their associated functional properties.
INTRODUCTION
Line 50 - I propose figures with appropriate resolution and unchanged dimensions (in width)
Line 66 - "... produce fragrance" - cannot be phrased in this way, the flowers contain volatile compounds which give off a characteristic odor.
MATERIALS AND METHODS
Line 377- More information about the plant material is needed. The amount of latex sample collected, how many plants latex was collected from, were these plants cultivated or was latex collected from wild species.
Line 384-385- Evaporation conditions (temperature and pressure) must be provided.
Line 386- Ultrasonic application conditions must be specified: Type of equipment (bath or probe system - manufacturer, country of origin), frequency, rated output power of the equipment, duration of treatment.
Author Response
Thank you for your comments to improve our manuscript: All our changes were highlighted with yellow color.
REVIEWER 1
The article provides new information on the composition of phytochemicals of the milky sap (latex) of the species Himatanthus sucuuba ( Spruce ), as well as an experiment on the effect of latex doses on cytotoxicity and the possibility of wound healing in mice. The UPLC-ECI- MS / MS technique also identifies compounds not previously identified in other literatures, providing important information on how the potential site, i.e. the specific pedoclimatic conditions in which the plants grew (Peru), play a role in phytochemicals composition in plant material, latex.
The introduction is clearly written and provides all the necessary facts about the research questions. The results and discussion are presented and explained clearly and concisely, and all tables and figures are necessary to present the results adequately. The conclusion is in line with the aim of the paper and clearly highlights the main findings, but also provides a critical review of the research that should be carried out as a follow-up.
In the materials of the paper, it is necessary to describe in more detail the origin of the plant material, but I give all the suggestions below.
Precisely because of all these points, after minor corrections according to the suggestions, I recommend the publication of the paper.
TITLE - I suggest a shorter, more concrete title, for example: Phytochemical screening of Himatanthus sucuuba (Spruce) Woodson (Apocynaceae) latex, in vitro cytotoxicity and incision wound repair in mice.
Reply 1: Thank you for you recommendation. We modifiedd it according to your proposed title.
ABSTRACT
Lines 21-22 - I propose for the first sentence a brief description of the phytochemicals of Himatanthus sucuuba and their associated functional properties.
Reply 2: Thank you for this point. We amended according to your comments. (Himatanthus sucuuba known as “Bellaco caspi” is a medicinal plant, which latex, stem bark or leaves possess phenolic acids, lupeol, β-dihydro-plumbericinic acid, plumericin, plumeride, among others. Some of them have been linked to biological activities such as antiulcer, anti-inflammatory, and wound healing.)
INTRODUCTION
Line 50 - I propose figures with appropriate resolution and unchanged dimensions (in width)
Reply 3: Thank you for this point. We amended figures and included those with high resolution.
Line 66 - "... produce fragrance" - cannot be phrased in this way, the flowers contain volatile compounds which give off a characteristic odor.
Reply 4: Thank you for this point. We amended according to your comments.
MATERIALS AND METHODS
Line 377- More information about the plant material is needed. The amount of latex sample collected, how many plants latex was collected from, were these plants cultivated or was latex collected from wild species.
Reply 5: Thank you for this point. We amended according to your comments.
Line 384-385- Evaporation conditions (temperature and pressure) must be provided.
Reply 6: Thank you for this point. We amended according to your comments. Buchi Rotavapor R-200, Switzerland, (40°C, 130 mbar).
Line 386- Ultrasonic application conditions must be specified: Type of equipment (bath or probe system - manufacturer, country of origin), frequency, rated output power of the equipment, duration of treatment.
Reply 7: Thank you for this point. We amended according to your comments. ultrasonicator (40 kHz, heat power 150 W; Branson 3800, USA) for 30 min.

Reviewer 2 Report
1- In the cytotoxicity test, are the authors studied the effect of 1% DMSO? As the used dose in the study is very high.
2- The authors focused the cytotoxic activity of H. sucuuba latex on colon cancer cell lines only. I think other types of cancer cell lines should be used.
3- The authors should be detecting the mechanism of action of H. sucuuba latex on cancer cells which may be through inhibition of cancer cell proliferation or cell apoptosis.
4- In wound healing of H. sucuuba latex in mice, what is the used dose injected in mice and type of the injection.
5- What is the positive control used in wound healing of H. sucuuba latex in mice?
6- The authors must study the in vitro effect of H. sucuuba latex against different types of cancer cells through using cell migration assay.
7- Positive control should be used in wound healing of H. sucuuba latex either in vivo or in vitro.
Author Response
Dear Editor In-Chief
Thank you for your comments to improve our manuscript. We have amended our manuscript point by point according to comments and suggestions of reviewers. All changes were highlighted with yellow color.
1- In the cytotoxicity test, are the authors studied the effect of 1% DMSO? As the used dose in the study is very high.
Reply 1: Thank you for your observations, we did study the effect of 1% DMSO but we considered unnecessary to report values of 1% DMSO in our table. We used different concentrations (5–1000 µg/ml) for latex because we do not have any previous reported work with H. sucuuba latex on colon cancer cells.
2- The authors focused the cytotoxic activity of H. sucuuba latex on colon cancer cell lines only. I think other types of cancer cell lines should be used.
Reply 2: Thank you for your observations. In effect, we only used two gastrointestinal cells based on our antecedents of H. sucuuba used for ulcers and gastrointestinal cancer, furthermore, we worked with two normal cells CHO-K1 and HaCat cell. For our profile on colon cancer only is absent HT-29 cells. Inhibition of SW480 and SW620 could reveal any effect on cancer cells in vitro. However, we need to study in a in vivo model in the future.
3- The authors should be detecting the mechanism of action of H. sucuuba latex on cancer cells which may be through inhibition of cancer cell proliferation or cell apoptosis.
Reply 3: Thank you for your observations. We did not work additional tests to demonstrate cell proliferation or cell apoptosis. Our group will consider develop additional assays in-vitro and publish a second paper with other types of tumor cells and establish any mechanism with any biochemical marker.
4- In wound healing of H. sucuuba latex in mice, what is the used dose injected in mice and type of the injection.
Reply 4. Thank you for your question. As we explained in the methodology, we used the latex in its natural form (100%) without any dilution according to its ethnopharmacological use. Fresh latex was collected and stored immediately at 4°C until further use. One milliliter of latex was applied by topical administration directly on the wound areas induced in mice. Some points were corrected and added in our methodology. Pag 16 (lines 419-421).
5- What is the positive control used in wound healing of H. sucuuba latex in mice?
Reply 5. Thank you for your observtion. We did not use a commercial drug according to our previous work and it was explained in the seven question. We used one group, who only received distilled water on the wounds to compare the latex group.
6- The authors must study the in vitro effect of H. sucuuba latex against different types of cancer cells through using cell migration assay.
Reply 6: Thank you for your observations. We considered to study the cytotoxic effect on other types of tumor cells and publish in a second version of our article, but we want to demonstrate a cytotoxic profile on colon cancer cells with this publication.
7- Positive control should be used in wound healing of H. sucuuba latex either in vivo or in vitro.
Reply 7: Thank you for your observations. In effect, we worked with two groups, but we considered unnecessary to add other group based on our previous published work, which we worked with one third group and received ZnO cream, furthermore we included in the discussion those reports and was referenced. Calero-Armijos LL, Herrera-Calderon O, Arroyo-Acevedo JL, Rojas-Armas JP, Hañari-Quispe RD, Figueroa-Salvador L. Histopathological evaluation of latex of Bellaco-Caspi, Himatanthus sucuuba (Spruce) Woodson on wound healing effect in BALB/C mice. Vet World. 2020;13(6):1045-1049. doi:10.14202/vetworld.2020.1045-1049
The difference with this experimental design is that we studied the wound-healing effect in 10 days and the latex was applied twice a day.
Author thanks reviewer, for its valuable efforts to improve our manuscript.
Reviewer 3 Report
The manuscript of Herrera-Calderón et al. described a phytochemical profiling of the latex of the Peruvian tree Himatanthus sucuuba on behalf of LC-MS/MS methods. Furthermore, cytotoxicity and woundhealing studies were performed on tumor cell lines, as well as by animal (mice) experiments. Via an in-silico docking study on glycogen synthase kinase 3 (GSK3), the authors have tried to model the protein binding side of several identified latex constituents.
Even the topic is interesting the manuscript have serious shortcomings in expression, grammar and spelling. Through this, the meanings of many sentences does not open up for the reader.
Consequently, the manuscript can not be published in the current version. It requires a major revision and correction by a native speaker. Also the structure of the manuscript have to be more concise.
Technical and general remarks:
The term „phytochemicals“ is a slang. It may partially substituted by terms like: natural constituents, phytochemical constituents or chemical compounds, e.c.
In the abstract a hint to the molecular modelling study is missing. Please complete!
Lines 41-42: Please specify what kind of plants are belonging to the Himatanthus species – trees, shrubs or herbal plants?
Lines 46-49: The species names can be abbreviate, like: H. drasticus, H. phagedaenicus, H. attenuatus … e.c.
Lines: 50-61: Pictures are distorded – please corrected!
Line 66-67: Its white flowers have a pleasant smell.
Table 1: It not makes sence to show molecular weights on four decimal points (I guess you not use high-resolution mass spectrometry?) – Specify to one decimal points, but the theoretical molecular weights (for the sum formulas) on two decimal points.
Table 2: What means selctivity index (SI)?
The chapter 2.4 appears very sophisticated but the sense does not open up for the general reader. You may add some additional remarks to the working method of this molecular modelling.
Lines 322 – 350: Shorten up or transfer some pictures to the supplementary material! You may show only one picture of the acacetin-7-O-neohesperidoside with the lowest binding energy?
Finally, a conclusion or summary is missing.
Author Response
Dear Editor In-Chief
Thank you for your comments to improve our manuscript. We have amended our manuscript point by point according to comments and suggestions of reviewers. All changes were highlighted with yellow color.
REVIEWER 3
The manuscript of Herrera-Calderón et al. described a phytochemical profiling of the latex of the Peruvian tree Himatanthus sucuuba on behalf of LC-MS/MS methods. Furthermore, cytotoxicity and wound healing studies were performed on tumor cell lines, as well as by animal (mice) experiments. Via an in-silico docking study on glycogen synthase kinase 3 (GSK3), the authors have tried to model the protein binding side of several identified latex constituents. Even the topic is interesting the manuscript have serious shortcomings in expression, grammar and spelling. Through this, the meanings of many sentences do not open up for the reader. Consequently, the manuscript cannot be published in the current version. It requires a major revision and correction by a native speaker. Also, the structure of the manuscript has to be more concise.
Technical and general remarks:
The term „phytochemicals“ is a slang. It may partially substituted by terms like natural constituents, phytochemical constituents or chemical compounds, e.c.
Reply 1: Thank you for your observation. Phytochemicals were changed by chemical compounds or phytochemical constituents
In the abstract a hint to the molecular modelling study is missing. Please complete!
Reply 2: Thank you for your observation. Molecular docking was added in the abstract. (line 30, pag 1)
Lines 41-42: Please specify what kind of plants are belonging to the Himatanthus species – trees, shrubs or herbal plants?
Reply 3: Thank you for your observation, Himatanthus species are generally trees and shrubs, both were referenced.
Lines 46-49: The species names can be abbreviated, like: H. drasticus, H. phagedaenicus, H. attenuatus … e.c.
Reply 4: Himatanthus species were abbreviated according to your comments.
Lines: 50-61: Pictures are distorded – please corrected!
Reply 5: Figures were ordered and included with high resolution. (Line 51 pag 2)
Line 66-67: Its white flowers have a pleasant smell.
Reply 6: This sentence was modified according to two reviewers.
Table 1: It not makes sence to show molecular weights on four decimal points (I guess you not use high-resolution mass spectrometry?) – Specify to one decimal points, but the theoretical molecular weights (for the sum formulas) on two decimal points.
Reply 7: Thank you for your observation, values of table 1 were corrected.
Table 2: What means selectivity index (SI)?
Reply 8: Thank you for your reply, it was included in caption of table 2 and methodology section 3.7. (pag 17, lines 438-441)
The chapter 2.4 appears very sophisticated, but the sense does not open up for the general reader. You may add some additional remarks to the working method of this molecular modelling.
Reply 9. Thank you for your observations, we corrected according to your suggestions in several points. (Pag 12, lines 284-292)
Lines 322 – 350: Shorten up or transfer some pictures to the supplementary material! You may show only one picture of the acacetin-7-O-neohesperidoside with the lowest binding energy?
Reply 10: We have re-ordered our figures and only included three compounds docked with high values in ranking order.
Finally, a conclusion or summary is missing.
Reply 11: Thank you for your observation, we included the conclusion in chapter 4. (Lines 477-492) pag 17. Additionally, expression, grammar and spelling were corrected and reviewed again.
Thank you for your valuable efforts to improve our manuscript.

Round 2
Reviewer 2 Report
1- DMSO is very toxic material and the authors used high percent (1%) therefore I am not agreeing with the authors as they replied ‘’ we considered unnecessary to report values of 1% DMSO. They must study the effect of DMSO on the viability of the used cells (cancerous and normal cells) then add these values in the manuscript.
2- The cytotoxic effect of H. sucuuba latex on HT-29 cancer cell line must be included in the study.
3- In vitro cell migration assay must be involved in the study to show the effect of H. sucuuba latex on the wound healing with using a reference control.
4- The results of wound healing of H. sucuuba latex in mice must be compared with reference control to validate the obtained results. Also, it is necessary that representative photos of mice's wounds must be added to the results section to show the wound before and after the treatment with H. sucuuba latex.
Author Response
1- DMSO is very toxic material and the authors used high percent (1%) therefore I am not agreeing with the authors as they replied ‘’ we considered unnecessary to report values of 1% DMSO. They must study the effect of DMSO on the viability of the used cells (cancerous and normal cells) then add these values in the manuscript.
R1: Thank you for your observation, DMSO is one of the solvents used for cytotoxicity assays, but its toxic effect is variable, and it is going to depend on the kind of cell lines tested for this purpose, such as this manuscript stated, which DMSO in low concentrations (0.1-1.5%) may interfere with important cellular processes as demonstrated in this study. Furthermore, this percentage range used is acceptable as it is referred here: Tunçer, S., Gurbanov, R., Sheraj, I. et al. Low dose dimethyl sulfoxide driven gross molecular changes have the potential to interfere with various cellular processes. Sci Rep 8, 14828 (2018). https://doi.org/10.1038/s41598-018-33234-z
In effect, our laboratory protocol uses 1% DMSO and there were not alterations or any influence to alter our results. Additional articles published by our research group has been accepted with this concentration of DMSO:
- Hernández C, Moreno G, Herrera-R A, Cardona-G W. New Hybrids Based on Curcumin and Resveratrol: Synthesis, Cytotoxicity and Antiproliferative Activity against Colorectal Cancer Cells. Molecules. 2021;26(9):2661. Published 2021 May 1 https://dx.doi.org/10.3390%2Fmolecules26092661
- Angie Herrera-R; Gustavo Moreno; Pedronel Araque; Isabel Vasquez; Elizabeth Naranjo; Fernando Alzate; Wilson Cardona-G. "In-vitro Chemopreventive Potential of a Chromone from Bomarea setacea (ALSTROEMERIACEAE) against Colorectal Cancer". Iranian Journal of Pharmaceutical Research, 20, 2, 2021, 254-267. http://ijpr.sbmu.ac.ir/article_1101427.html
According to your suggestions, we included these values in Table 2 and Table S1 in the supplementary material, where no changes and alterations were observed.
2- The cytotoxic effect of H. sucuuba latex on HT-29 cancer cell line must be included in the study.
R2. Thank you for your observations. We do not have HT-29 cancer cells in our laboratory by the moment. However, we considered that HT-29 is a model of less metastatic and more differentiated colon cancer cells. SW480 and SW620 both can be used as a model of more metastatic undifferentiated cancer cells. SW480 cells are associated to worse overall survival and higher risk of recurrence based on patient derived databases. A new assay with other types of tumor cell lines will be carried out in the future.
Christensen, J., El-Gebali, S., Natoli, M. et al. Defining new criteria for selection of cell-based intestinal models using publicly available databases. BMC Genomics 13, 274 (2012). https://doi.org/10.1186/1471-2164-13-274
3- In vitro cell migration assay must be involved in the study to show the effect of H. sucuuba latex on the wound healing with using a reference control.
R3. In effect, an in-vitro cell migration assay would complement our work, but we only designed an in-vivo model to demonstrate the wound healing effect in mice. The histopathological study confirms our hypothesis, according to findings revealed in figure . To establish any involved mechanism, in-vitro models will be tested in another study.
4- The results of wound healing of H. sucuuba latex in mice must be compared with reference control to validate the obtained results. Also, it is necessary that representative photos of mice's wounds must be added to the results section to show the wound before and after the treatment with H. sucuuba latex.
R4. Thank you for your observation. The ethical committee approved the experimental design of this protocol considering two groups because we worked with H. sucuuba latex in an experimental design very similar with our previous published report (Calero-Armijos LL, Herrera-Calderon O, Arroyo-Acevedo JL, Rojas-Armas JP, Hañari-Quispe RD, Figueroa-Salvador L. Histopathological evaluation of latex of Bellaco-Caspi, Himatanthus sucuuba (Spruce) Woodson on wound healing effect in BALB/C mice. Vet World. 2020;13(6):1045-1049. https://dx.doi.org/10.14202%2Fvetworld.2020.1045-1049).
In that article, which we referenced in this manuscript, we used ZnO as a positive control and non- significant differences were observed with latex. So, we only used two experimental groups, negative control and latex group. However, results obtained with ZnO were included in the discussion.
I am sorry but we did not record the macroscopic images to include it in this manuscript.
Reviewer 3 Report
The substantial quality of the manuscript Herrera-Calderón et al. Herrera-Calderón et al. is now clearly improved.
However, as I recommended in my first review, it‘s my impression that the manuscript have serious shortcomings in expression, syntax, grammar and spelling. So, from the linguistic point of view the manuscript have to be revised by a native speaker or professional translator!
It is not the reviewers job to do this.
Author Response
Dear Reviewer
Thank you for you observations.
POINT 1: The substantial quality of the manuscript Herrera-Calderón et al. H is now clearly improved. However, as I recommended in my first review, it‘s my impression that the manuscript have serious shortcomings in expression, syntax, grammar and spelling. So, from the linguistic point of view the manuscript have to be revised by a native speaker or professional translator!.
R1: Manuscript has been revised by a native speaker.
Round 3
Reviewer 2 Report
Although it is better to the authors to study the In vitro cell migration assay to strength the work, I recommend the acceptance of the manuscript.
Author Response
R1: Thank you for your recommendation, in effect , in-vitro cell migration assay will be carried in our next work with additional biochemical markers using in vitro assays.
I thank reviewer for recommending our manuscript to be published.